# Prognostic Role of Inflammatory and Nutritional Biomarkers in Non-Small-Cell Lung Cancer Patients Treated with Immune Checkpoint Inhibitors Alone or in Combination with Chemotherapy as First-Line

**DOI:** 10.3390/cancers16223871

**Published:** 2024-11-19

**Authors:** Antonello Veccia, Mariachiara Dipasquale, Stefania Kinspergher, Orazio Caffo

**Affiliations:** Medical Oncology, Santa Chiara Hospital, 38122 Trento, Italy; mariachiara.dipasquale@apss.tn.it (M.D.); stefania.kinspergher@apss.tn.it (S.K.); orazio.caffo@apss.tn.it (O.C.)

**Keywords:** prognostic, inflammatory, nutritional, biomarker, NSCLC, immune checkpoint inhibitors, chemotherapy

## Abstract

This study evaluated the prognostic role of the advanced lung cancer inflammation (ALI) index, lung immune prognostic index (LIPI), prognostic nutritional index (PNI), and systemic inflammation score (SIS) in 191 metastatic NSCLC patients receiving immunotherapy alone or in combination with chemotherapy as first-line at Santa Chiara Hospital of Trento, from 2017 to 2024. After a median follow-up of 27.7 months, a significantly longer OS was associated with an ALI score >18 vs. ≤18 (18.0 vs. 7.3 months; *p* = 0.00111), LIPI score 0 vs. 1 and 2 [18.9 vs. 8.2 and 4.2 months; (*p* = 0.001)], PNI ≥ 45 vs. <45 (22.7 vs. 9.6 months; *p* = 0.002), and SIS score 0 vs. 1 and 2 (27.4 vs. 7.1 and 8.6 months, respectively; *p* < 0.001). The study confirmed that the ALI, LIPI, PNI, and SIS represent useful tools to prognosticate survival in metastatic lung cancer patients treated with immunotherapy alone or in combination with chemotherapy.

## 1. Introduction

The therapeutic landscape of non-small-cell lung cancer (NSCLC) stage IV without driver alterations has completely changed since the introduction of immune checkpoint inhibitors (ICIs) into clinical practice [1,2]. Pivotal phase 3 trials have well established the role of programmed cell death protein 1 (PD-1) and programmed death ligand 1 (PD-L1) inhibitors as first-line treatment. These agents were used alone in the presence of PDL1 expression ≥ 50% [3,4] or in combination with chemotherapy ± cytotoxic T-lymphocyte-associated protein 4 (CTLA-4) inhibitors regardless of PDL1 expression levels [5,6,7,8].

Unfortunately, only a percentage of patients ranging from 18.5% to 31.9% showed a long-term response to treatment, as reported by pivotal trials with longer follow-up [9,10,11], while most of them showed primary or secondary resistance [12,13]. Moreover, some patients may develop hyperprogression [14] or manifest severe immune-related adverse events (irAEs), which can also be life-threatening [15]. Therefore, it is important to select biomarkers that can identify patients who can benefit most from ICI treatment, while avoiding the risk of unnecessary toxicities for others. Currently, apart from PDL1 used to select patients eligible for ICI as single agent or for ICI-based combinations [16], no biomarker can guide the choice of systemic treatment in patients with advanced NSCLC.

In recent years, several studies have evaluated inflammation-related factors as potential biomarkers of systemic inflammation in NSCLC patients treated with ICI. These factors include C-reactive protein (CRP), lactate dehydrogenase (LDH), circulating white blood cells (WBC), absolute neutrophil count (ANC), neutrophil-to-lymphocyte ratio (NLR), derived NLR, and platelet-to-lymphocyte ratio (PLR) [17]. Similarly, nutritional parameters, such as albumin, have been evaluated for their potential prognostic and predictive role during ICI treatment [18].

By combining these factors, different prognostic scores have been developed. The most promising are the lung immune prognostic index (LIPI) [19], the systemic inflammation score (SIS) [20], the advanced lung cancer inflammation (ALI) index [21], and the prognostic nutritional index (PNI) [22].

The aim of this study was to retrospectively confirm the prognostic role of the LIPI, SIS, ALI, and PNI in patients with metastatic NSCLC undergoing first-line treatment with ICI alone or in combination with chemotherapy. Moreover, we investigated whether one biomarker could prognosticate survival better than others.

## 2. Patients and Methods

### 2.1. Study Population and Data Collection on Disease and Treatment

We conducted a retrospective study including a consecutive series of patients with histologically diagnosed metastatic NSCLC, which had received at least one course of immune checkpoint inhibitor alone or in combination with chemotherapy as first-line treatment at the Santa Chiara Hospital of Trento, Italy, from January 2017 to March 2024. Patients with PDL1 ≥ 50% could receive anti-PD1 (pembrolizumab) or anti-PDL1 (atezolizumab) monotherapy, while those with PDL1 < 50% could receive pembrolizumab + chemotherapy or ipilimumab + nivolumab + chemotherapy combinations, according to daily clinical practice. A minimum follow-up of 3 months was required. Biomolecular analyses were performed using the diagnostic methods available at our Institute (Sequenom, real-time PCR, next-generation sequencing), while PDL1 expression levels were analyzed in tumor cells by immunohistochemistry, according to the assay currently used.

We collected the following baseline characteristics of patients from the clinical records: sex, weight and height, body mass index (BMI), date of metastatic disease diagnosis, age at diagnosis, smoking status, number of comorbidities (cardiovascular, pulmonary, endocrine, hematological, rheumatological, neurological, cutaneous, ocular, gastrointestinal, or psychiatric), ECOG performance status, histologic subtype, stage (according to the 8th TNM edition), number and type of metastatic sites, and biomolecular phenotype, including PDL1 expression levels and mutational status of EGFR/ALK/ROS1/KRAS/BRAF/other genes (MET, RET, HER2, PIK3CA). The following data on ICI-based treatment were recorded: type of ICI, any chemotherapeutic agents administered concomitantly, date of treatment start, best response to the treatment, date and reason of discontinuation, duration of the treatment, palliative radiotherapy treatments, and subsequent lines of treatment. Imaging monitoring was performed according to the local clinical practice. The response was evaluated according to Response Evaluation Criteria in Solid Tumors (RECIST) version 1.1.

Finally, vital status (alive or dead) and date of death/last follow-up were collected.

The local ethical committee gave its approval.

### 2.2. Laboratory Tests and Biomarkers

Pretreatment laboratory exams were retrieved from the patients’ medical records: blood count including both total white blood cells (WBC) and their subpopulations (neutrophils, lymphocytes, eosinophils), hemoglobin, platelets, albumin, calcium, and lactate dehydrogenase (LDH). They were reported according to the limits of the local laboratory, had to be performed within 14 days before starting ICI-based treatment, and were used to calculate the following predefined inflammatory and nutritional biomarkers:

NLR = neutrophils × 10^9^/lymphocytes × 10^9^;

dNLR = neutrophils × 10^9^/(white blood cells × 10^9^ − neutrophils × 10^9^);

LIPI score, based on dNLR (ratio ≤ 3 and >3 were scored 0 and 1, respectively) and LDH (score 0 and 1 indicated values ≤ and >241 U/L, respectively); the LIPI score defined three prognostic groups: good (0 factors), intermediate (1 factor), and poor (2 factors);

Systemic inflammatory score (SIS), calculated as the sum of the following factors, scored one point each: hemoglobin < 12.5 g/dL and serum albumin < 3.6 g/dL, resulting in scores of 0–2;

Advanced lung cancer inflammation index (ALI): (serum albumin × BMI)/NLR; BMI = weight (kg)/height (m)^2^;

PNI, calculated as the sum of albumin (g/dL) + 0.005 × lymphocytes (10^9^).

Cut-offs for albumin (<3.5 g/dL), NLR (≤5, >5), ALI score (≤18, >18), and PNI (<45, ≥45) were based on previous studies examining these factors and not derived from the present analysis.

### 2.3. Statistical Analysis

Descriptive statistics were used to report patients’ characteristics: median with interquartile range was used to describe continuous variables, and frequency (percentage) for categorical variables.

OS was calculated from the start of the ICI-based treatment until death due to any cause or the date of the last follow-up for censored patients. PFS was calculated from the start of the ICI-based treatment until disease progression or death due to any cause or the date of the last follow-up for censored patients. The duration of the treatment was calculated from the start of ICI-based treatment until treatment discontinuation or death.

Disease control rate (DCR) was defined as the sum of complete response rate, partial response rate, and stable disease rate.

Kaplan–Meier survival curves were used to estimate median OS and PFS including 95% confidence interval (95% CI) according to different prognostic scores. Differences were tested via the log-rank test.

A Cox proportional hazards model was used to develop multivariable prediction models for OS and PFS. A backward variable selection method with a type I error criterion of 0.05 was used to select factors significantly affecting PFS and OS.

The c-statistic was used to study discrimination, which refers to the ability of prognostic biomarkers to assign higher predicted risks to subjects who died during the follow-up than to subjects who survived during the follow-up period. The discrimination was also assessed by using the survival receiver operating characteristic (ROC) curves, which evaluate sensitivity and specificity of prognostic biomarkers to predict survival. The predictive power of response for each biomarker was also tested by ROC curves analysis. Statistical analyses were performed with R software (version 4.4.1) [23].

## 3. Results

Baseline characteristics of the 191 patients included in the study are shown in Table 1. Most of patients were male (65.4%) with a median age of 70 years. The most frequent histotype was adenocarcinoma (75.4%), and about one-third (33%) of patients were PDL1-negative, while nearly half had PDL1 ≥ 50%. All patients had a diagnosis of metastatic NSCLC, with liver and brain metastases in 14.1% and 21.5% of them, respectively. All patients received first-line systemic treatment based on ICI: 93 patients received ICI alone and 98 patients were treated with a combination of chemo + ICI. No statistically significant differences were found between the baseline characteristics of the two groups, except for PDL1 expression levels and consequently the type of treatment: patients in the ICI group received pembrolizumab (87.1%) or atezolizumab (12.9%), while those in the chemo + ICI group received pembrolizumab + chemotherapy (99%) or ipilimumab + nivolumab + chemotherapy (1%). In addition, there was a higher proportion of patients with mutant KRAS in the ICI subgroup than in the chemo + ICI subgroup (38.7% vs. 26.5%; *p* = 0.042) (Appendix A).

The median follow-up was 27.7 (range 0.4–74.2) months. The median duration of the treatment was 4.7 (range 0.27–78.7) months.

Cox proportional regression analysis included patient-related factors (age at diagnosis, ECOG PS, comorbidities, pretreatment albumin), disease-related factors (histology, brain metastases, liver metastases), treatment-related factors (mono vs. combo immunotherapy, drugs), and prognostic indices (ALI, LIPI, PNI, SIS).

In Cox proportional regression analysis, pretreatment albumin as a continuous variable was significantly associated with better OS. Patients with ECOG PS 1 showed a worse PFS (HR 2.43; 95% CI 1.63–3.62; *p* < 0.0001) and OS (HR 2.38; 95 CI 1.58–3.59; *p* < 0.0001). The presence of liver metastases was also significantly associated with worse PFS and OS. None of the indices (ALI, PNI, LIPI, SIS) was found to be an independent prognostic factor: only LIPI = 1 showed a statistically significant association with worse PFS (Table 2).

The DCR was 61.2% in the overall population, with no significant differences between ICI and chemo + ICI groups (57% vs. 65.3%; *p* = 0.151) (Table 1).

Median OS by prognostic score (ALI, LIPI, PNI, and SIS) in the overall population and stratified by treatment (ICI vs. ICI + chemo) is shown in Table 3.

The median OS was 7.3 (95% CI 4.8–10.6) months in the ALI score ≤ 18 group vs. 18.0 (95 CI 13.0–28.3) months in the ALI score > 18 group (*p* = 0.001) (Figure 1). The statistically significant advantage in OS was maintained by stratifying by treatment (ICI vs. ICI + chemo) (Table 3).

The median OS was 18.9 (14.4–30.9), 8.2 (6–14.6), and 4.2 (2–5-9.7) months in the good, intermediate, and poor prognosis LIPI groups, respectively (*p* = 0.001) (Figure 2). The statistically significant advantage in OS was maintained by stratifying by treatment (ICI vs. ICI + chemo) (Table 3).

The median OS was 9.6 (6.6–13.2) months in the PNI < 45 group vs. 22.7 (13–36.7) months in the PNI ≥ 45 group (*p* = 0.002) (Figure 3). Also for this score, the statistically significant advantage in OS was maintained by stratifying by treatment (ICI vs. ICI + chemo) (Table 3).

The median OS was 27.4 (17.4–36.7), 7.1 (5–12.6), and 8.6 (4.3–13.2) months in the SIS 0, 1, and 2 groups, respectively (<0.001) (Figure 4). The statistically significant advantage in OS was maintained by stratifying by treatment (ICI vs. ICI + chemo) (Table 3).

Model discriminations (c-statistic) of the ALI, LIPI, PNI, and SIS for OS were estimated to be 0.638 (95% CI 0.553–0.723), 0.577 (95% CI 0.5–0.653), 0.626 (95% CI 0.541–0.71), and 0.593 (95% CI 0.509–0.676), respectively. Even combining the four indices, the performance of the model did not improve: the AUC was 0.634 (95% CI 0.55–0.718). Furthermore, the four indices showed poor predictive power of response: the AUC of ALI, LIPI, PNI, and SIS was estimated to be 0.505 (95% CI 0.389–0.62), 0.527 (95 CI 0.432–0.622), 0.492 (95% CI 0.377–0.606), and 0.488 (95% CI 0.388–0.588), respectively.

## 4. Discussion

In recent years, several clinical, laboratory, and genetic biomarkers, most of which are easily obtained in daily clinical practice, have been investigated [17] to better select patients for ICI-based treatments. Inflammatory and nutritional biomarkers are the most promising. Indeed, inflammation is associated with cancer growth, metastasis, and spread [24]; moreover, it is an unfavorable prognostic factor, typically related to malnutrition, hypoalbuminemia, weight loss, and cancer cachexia.

In the current work, we found that inflammatory and nutritional biomarkers such as the ALI, LIPI, PNI, and SIS might prognosticate OS in patients with metastatic NSCLC receiving ICI alone or in combination with chemotherapy as first-line treatment.

Patients with an ALI score > 18 lived significantly longer than those with an ALI score ≤ 18, regardless of the addition of chemotherapy to ICI treatment.

These data on the ALI index, already known since 2013 for its prognostic role in lung cancer patients treated with chemotherapy [25], are in line with those reported by a recent retrospective study including 672 patients with stage IV NSCLC treated with PD-L1 inhibitors alone or in combination with chemotherapy. In this study, the ALI confirmed its stronger predictive effect than NLR, PD-L1 tumor proportion score, lung immune prognostic index, and EPSILoN scores [21].

In 2018, Mezquita reported that the pretreatment LIPI, combining dNLR and LDH, was related to worse outcomes for ICI, but not for chemotherapy [19]. We found that the LIPI may also be a prognostic biomarker in patients undergoing chemoimmunotherapy.

The LIPI and PNI were found to predict survival outcomes in patients with advanced NSCLC treated with chemoimmunotherapy, particularly in patients with PD-L1 TPS < 50% [22]. A meta-analysis of 10 studies including more than 5000 patients with lung cancer reported that a low PNI correlated with unfavorable OS (HR = 1.72; 95% CI, 1.43–2.06; *p* = 0.000), confirming the prognostic role of the PNI [26]. These data were also confirmed by a recent meta-analysis [27].

In a retrospective study, pretreatment PNI was found to be an independent prognostic factor for OS in advanced NSCLC patients receiving pembrolizumab alone or chemoimmunotherapy as first-line treatment (*p* = 0.0270 and 0.0006, respectively) [28]. These data were confirmed by the results of our study: PNI ≥ 45 was associated with longer OS in patients treated with ICI as well as in those treated with combinations.

We also reported that SIS score could prognosticate OS, confirming its role as a marker of ICI’s efficacy, as reported by a previous study including patients with inoperable lung cancer and treated with first-line monoimmunotherapy (atezolizumab, nivolumab or pembrolizumab) [20].

Through multivariate analysis, we found that pretreatment albumin was significantly associated with OS.

Serum albumin is a biomarker of general health status, usually classified as low or normal using the cutoff of 3.5 g/dL. Cancer patients usually have hypoalbuminemia, associated with increased turnover: this is due to an inflammatory state rather than poor nutritional status, decreased synthesis, or protein loss through excretion [18]. Hypoalbuminemia is a recognized poor prognostic factor in cancer. In a single-center retrospective study of 210 advanced NSCLC patients receiving ICI with or without chemotherapy as first-line therapy, a significant association of OS with pretreatment albumin and its early decrease was found during treatment with ICI monotherapy but not with chemoimmunotherapy [29].

Upon multivariate analysis, ECOG PS 1 and the presence of liver metastases at diagnosis were associated with worse OS. Patients with worse ECOG PS are probably frailer and respond less to immunotherapy-based treatment, such as those with liver metastases at diagnosis, which is a negative prognostic factor in advanced NSCLC. Interestingly, in our study, the proportion of patients with liver metastases (14.1%) was higher than that reported in phase 3 randomized clinical trials and much closer to data from real-world settings.

Data from the literature show that other prognostic biomarkers were retrospectively investigated in metastatic NSCLC patients treated with first-line pembrolizumab plus platinum-based chemotherapy: the NHS lung prognostic score [30] and the Scottish inflammatory prognostic score (SIPS) [31] have been shown to predict survival. Instead, the pretreatment Gustave Roussy Immune (GRIm) score, which takes into account NLR, albumin, and LDH, did not show a prognostic role in patients who received pembrolizumab in the first-line setting for advanced NSCLC patients with PDL1 ≥ 50% [32].

The prognostic role of inflammatory and nutritional indices was also investigated in the preoperative setting, in NSCLC patients undergoing surgery [33], and in second-line and beyond, such as the EPSILoN score [34] and the Glasgow prognostic score (GPS) [35].

In our study, we tested the c-statistic of the different biomarkers to identify which of them might be preferred. Upon ROC curve analysis, the AUCs for OS were 0.577 and 0.593 for the two inflammatory biomarkers, the LIPI and SIS, respectively, while they were 0.638 and 0.626 for the two nutritional biomarkers, the ALI and PNI, respectively. Although discrimination tests suggested that all biomarkers had a moderate power as prognostic survival models, the higher c-statistic of the ALI and PNI suggests that scores including both metabolic parameters (such as albumin and BMI) and blood count cells (such as lymphocytes and NLR) are more complete and perform better. Even combining the four indices, the performance of the model did not improve.

These biomarkers alone are probably not sufficient to predict the survival of patients undergoing ICI-based treatment. We should combine them with clinical, histological, biomolecular, and radiological factors to increase the power of future prognostic models.

Moreover, none of the four biomarkers was found to be predictive of response. However, this topic should be further investigated: in fact, biomarkers such as LDH, NLR, or albumin reflect a certain inflammatory and nutritional status of the patient before starting immunotherapy and, therefore, they could influence the response to treatment.

A recent study explored the relationship between tumor metabolic glycolysis and inflammatory or nutritional status in advanced NSCLC patients receiving ICI alone or in combination with chemotherapy. An 18F-FDG PET imaging was performed before the initial PD-1 blockade, and the following scores were collected: NLR, PLR, SII, PNI, ALI, and GPS. High metabolic tumor volume, associated with high PLR and SII and low ALI, was identified as a significant factor for predicting shorter OS after first-line PD-1 blockade [36]. This confirms the interest of nuclear medicine in metabolic and inflammatory biomarkers.

Some limitations of our study need to be highlighted. Firstly, the analysis was retrospective and single-center, resulting in the risk of selection bias. Secondly, an external validation cohort was lacking. Thirdly, only pretreatment biomarkers were analyzed for their prognostic impact on survival, without longitudinal assessment to correlate with the subsequent therapies received by patients along their course of care. In fact, recent data suggest that chemotherapy and radiotherapy may alter the tumor microenvironment, including immune cells and soluble biochemical factors [37,38], modifying biomarkers such as those analyzed. Finally, the thresholds of the tested biomarkers were derived from previous studies, further making comparisons difficult.

## 5. Conclusions

Inflammatory and nutritional biomarkers such as the ALI, LIPI, PNI, and SIS represent useful tools to prognosticate survival in metastatic lung cancer patients treated with ICI-based schedules. They are easily calculated and inexpensive, so they could be used in conjunction with clinical assessments to provide additional objective information. Future studies should prospectively validate them to integrate their use into daily decision-making.

## Figures and Tables

**Figure 1 cancers-16-03871-f001:**
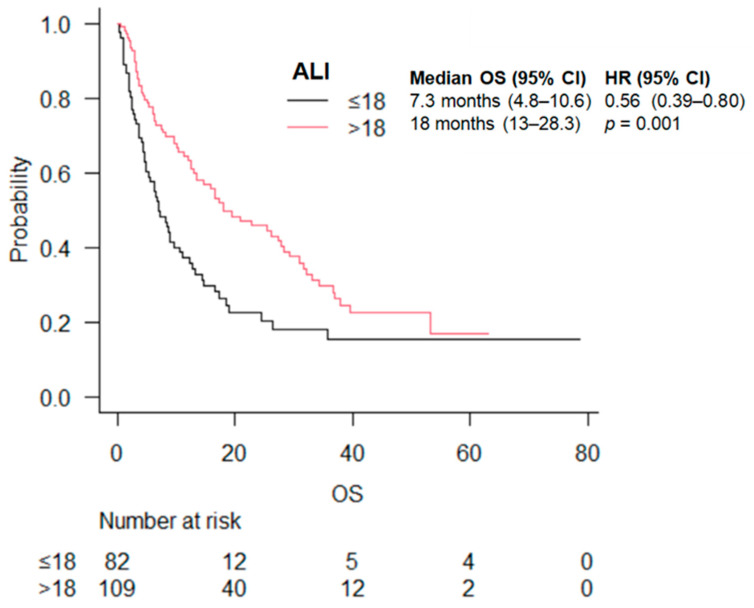
OS according to ALI score.

**Figure 2 cancers-16-03871-f002:**
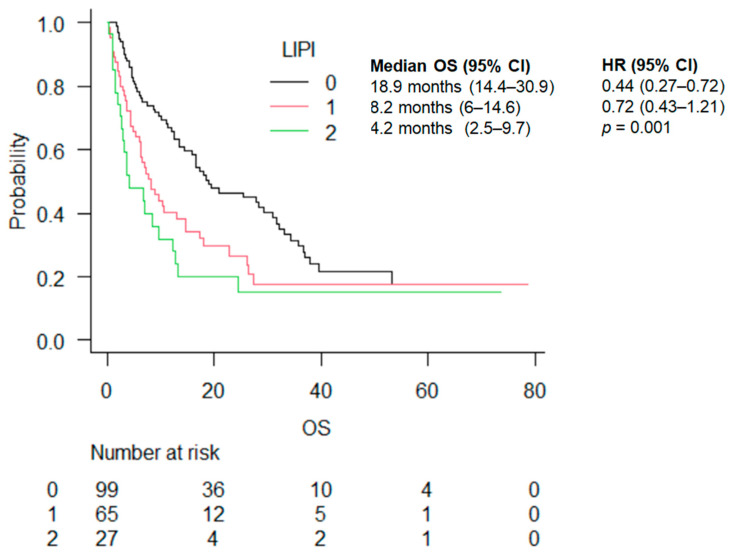
OS according to LIPI score.

**Figure 3 cancers-16-03871-f003:**
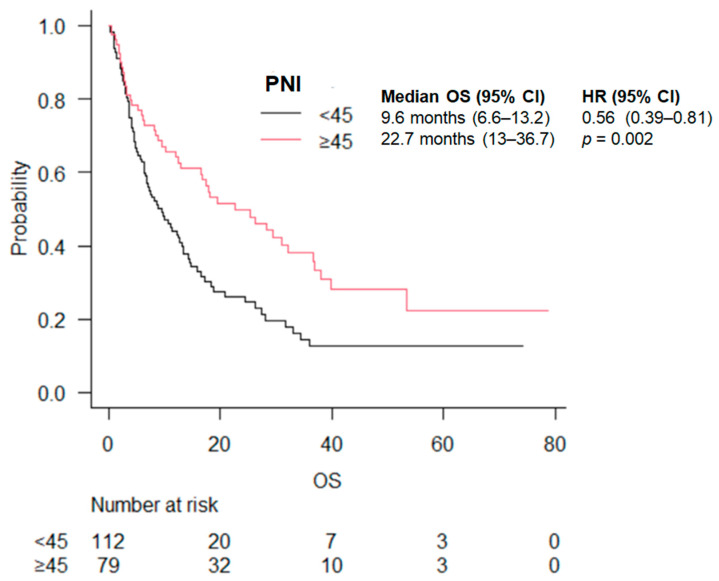
OS according to PNI score.

**Figure 4 cancers-16-03871-f004:**
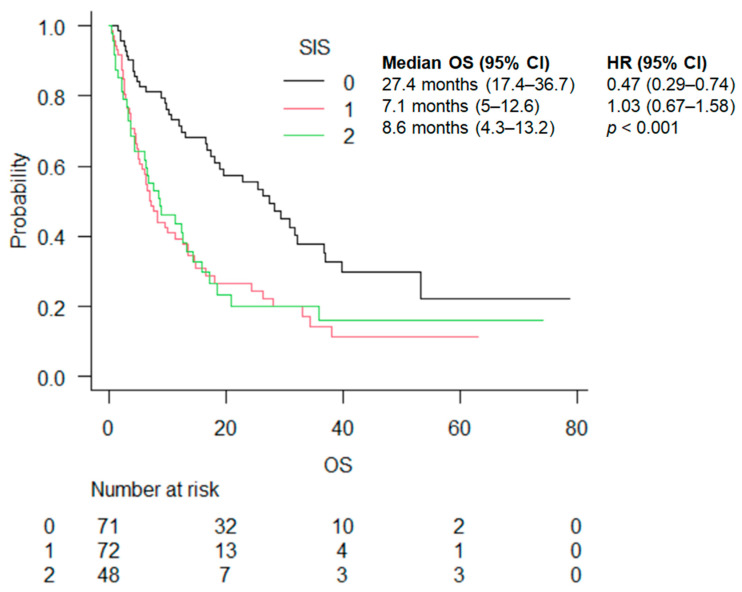
OS according to SIS score.

**Table 1 cancers-16-03871-t001:** Patients’ characteristics.

		All	ICI	ICI + chemo	*p* Value
		*n* (%)	*n* (%)	*n* (%)	
		191 (100)	93 (48.7)	98 (51.3)	
**Age at diagnosis, m (IQr)**		68.98 (64.75, 74.70)	70.41 (63.92, 76.46)	69.04 (64.96, 73.45)	0.075
**Sex, n (%)**	**M**	125 (65.4)	59 (63.4)	66 (67.3)	0.648
	**F**	66 (34.6)	34 (36.6)	32 (32.7)	
**ECOG PS, n (%)**	**0**	139 (72.8)	68 (73.1)	71 (72.4)	0.924
	**1**	44 (22.5)	20 (21.5)	23 (23.5)	
	**2**	8 (4.2)	4 (4.3)	4 (4.1)	
	**3**	1 (0.5)	1 (1.1)	0 (0)	
**Smoking, n (%)**	**never**	18 (9.4)	5 (5.4)	13 (13.3)	0.107
	**former**	92 (48.2)	49 (52.7)	43 (43.9)	
	**smoker**	70 (36.6)	36 (38.7)	34 (34.7)	
	**unknown**	11 (5.8)	3 (3.2)	8 (8.2)	
**Comorbidities, m (IQr)**		2 (1.3)	2 (1.3)	2 (1.3)	0.130
**Albumin, m (IQr)**		3.50 (3.00, 3.80)	3.50 (3.20, 3.90)	3.40 (2.92, 3.77)	0.154
**Histology, n (%)**	**adenocarcinoma**	140 (75.4)	66 (71.0)	78 (79.6)	0.085
	**squamous carcinoma**	44 (20.9)	25 (26.9)	15 (15.3)	
	**others**	7 (3.7)	2 (2.2)	5 (5.1)	
**PDL1, n (%)**	**≤1%**	63 (33.0)	0 (0)	63 (64.3)	<0.001
	**1–49**	34 (17.8)	0 (0)	34 (34.7)	
	**≥50%**	93 (48.7)	93 (100.0)	0 (0)	
	**not determinable**	1 (0.5)	0 (0)	1 (1.0)	
**Stage, n (%)**	**IV**	191 (100)	93 (100)	98 (100)	
**Brain mets, n (%)**	**no**	150 (78.5)	72 (77.4)	78 (79.6)	0.728
	**yes**	41 (21.5)	21 (22.6)	20 (20.4)	
**Liver, n (%)**	**no**	164 (85.9)	78 (83.9)	86 (87.8)	0.534
	**yes**	27 (14.1)	15 (16.1)	12 (12.2)	
**Line, n (%)**	**I**	191 (100)	93 (100)	98 (100)	
**Drug, n (%)**	**atezolizumab**	12 (6.3)	12 (12.9)	0 (0)	<0.001
	**pembrolizumab**	178 (93.2)	81 (87.1)	97 (99.0)	
	**IPI + NIVO**	1 (0.5)	0 (0)	1 (1.0)	
**ALI score, n (%)**	**≤18**	82 (42.9)	41 (44.1)	41 (41.8)	0.772
	**>18**	109 (57.01)	52 (55.9)	57 (58.2)	
**LIPI, n (%)**	**0**	99 (51.8)	53 (57.0)	46 (46.9)	0.367
	**1**	65 (34.0)	29 (31.2)	36 (36.7)	
	**2**	27 (14.1)	11 (11.8)	16 (16.3)	
**PNI, n (%)**	**<45**	112 (58.6)	48 (51.6)	64 (65.3)	0.058
	**≥45**	79 (41.4)	45 (48.4)	34 (34.7)	
**SIS, n (%)**	**0**	71 (37.2)	35 (37.6)	36 (36.7)	0.916
	**1**	72 (37.7)	36 (38.7)	36 (36.7)	
	**2**	48 (25.1)	22 (23.7)	26 (26.5)	

n = number; m = median; IQr = interquartile range.

**Table 2 cancers-16-03871-t002:** Multivariate analysis for PFS and OS.

	PFS		OS	
Variable	HR (95% CI)	*p*-Value	HR (95% CI)	*p*-Value
**Pretreatment albumin**	0.93 (0.88–0.97)	0.001	0.93 (0.89–0.98)	<0.001
**PS.ECOG**				
0	1		1	
1	2.43 (1.63–3.62)	<0.0001	2.38 (1.58–3.59)	<0.0001
2	1.77 (0.76–4.08)	0.179	1.77 (0.75–4.15)	0.190
3	0 (0-inf)	0.993	0 (0–inf)	0.994
**Liver**				
no	1		1	
yes	1.99 (1.21–3.29)	0.006	1.79 (1.09–2.95)	0.021
**LIPI**				
**0**	1		1	
**1**	1.66 (1.11–2.48)	0.012	1.77 (1.17–2.68)	0.006
**2**	1.85 (1.06–3.20)	0.027	1.61 (0.93–2.78)	0.083
**SIS**				
**0**	1		1	
**1**	1.39 (0.83–2.32)	0.199	1.42 (0.85–2.38)	0.178
**2**	0.63 (0.30–1.31)	0.223	0.87 (0.42–1.81)	0.714

Hazard ratio with 95% CI and *p*-values obtained from Cox regression model.

**Table 3 cancers-16-03871-t003:** OS according to prognostic scores and type of treatment.

		All	HR	*p* Value	ICI	HR	*p* Value	ICI + Chemo	HR	*p* Value
		Month (95% CI)	95% CI		Month (95% CI)	95% CI		Month (95% CI)	95% CI	
**Prognostic score**										
**ALI**	**≤18**	7.3 (4.8–10.6)	0.56 (0.39–0.80)	0.001	8.2 (4.5–14.6)	0.69 (0.42–1.12)	0.1	6.9 (4.2–9.7)	0.40 (0.23–0.67)	0.001
	**>18**	18.0 (13.0–28.3)			20.8 (12.4–33.1)			15.9 (11.4–28.3)		
**LIPI**	**0**	18.9 (14.4–30.9)	0.44 (0.27–0.72)	0.001	18.9 (12.4–33.1)	0.58 (0.28–1.20)	0.2	16.7 (12–31.7)	0.31 (0.16–0.62)	<0.001
	**1**	8.2 (6–14.6)	0.72 (0.43–1.21)		8.2 (3.27–22.7)	0.80 (0.36–1.75)		7.7 (5.3–14.7)	0.64 (0.33–1.27)	
	**2**	4.2 (2–5-9.7)			6.8 (1.1–24.4)			4 (2.5–12.3)		
**PNI**	**<45**	9.6 (6.6–13.2)	0.56 (0.39–0.81)	0.002	8.2 (4.6–17)	0.62 (0.38–1.01)	0.05	9.6 (6.3–13.5)	0.51 (0.29–0.90)	0.002
	**≥45**	22.7 (13–36.7)			22.7 (8.4–36.9)			18.1 (10.3–NA)		
**SIS**	**0**	27.4 (17.4–36.7)	0.47 (0.29–0.74)	<0.001	31 (16.6–39.7)	0.79 (0.40–1.58)	0.001	27.4 (12–31.7)	0.24 (0.12–0.47)	<0.001
	**1**	7.1 (5–12.6)	1.03 (0.67–1.58)		5.1 (2.5–8.3)	2.10 (1.01–4.02)		10 (6–16.6)	0.44 (0.24–0.81)	
	**2**	8.6 (4.3–13.2)			18.4 (6.3–NA)			6 (2.9–11.4)		

## Data Availability

The dataset generated and/or analyzed during the current study is available from the corresponding author on reasonable request.

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
