# Peer review of "Prognostic Role of Inflammatory and Nutritional Biomarkers in Non-Small-Cell Lung Cancer Patients Treated with Immune Checkpoint Inhibitors Alone or in Combination with Chemotherapy as First-Line"

_cancers, 2024, doi:10.3390/cancers16223871_

Round 1

Reviewer 1 Report (Previous Reviewer 1)

Comments and Suggestions for Authors

The authors took into account the reviewer's comments made at the previous stage of reviewing and substantially revised the manuscript. I believe that in its current form the manuscript can be recommended for publication.

Author Response

We thank the reviewer. 

Reviewer 2 Report (Previous Reviewer 2)

Comments and Suggestions for Authors

The authors properly responded to my comments. The English editing is recommended to improve the general readability of text and sentence flow.

Author Response

We thank the reviewer for the suggestion; the English was improved. 

This manuscript is a resubmission of an earlier submission. The following is a list of the peer review reports and author responses from that submission.

Round 1

Reviewer 1 Report

Comments and Suggestions for Authors

1. Table 1 is somewhat unsystematic: it should be grouped differently, for example, first gender, age, BMI, stage, then therapy, line of therapy, ..., group the indices or even separate them into a separate table. Metastases in one place, concomitant pathologies too (which ones, by the way, please provide a list)

2. There are no treatment factors in the multivariate analysis in Table 2? Line of therapy and drugs do not affect survival rates?

3. Why are only median survival rates given in Table 3? The Hazard ratio should be provided. It is necessary to conduct a multivariate analysis of factors from Tables 2 and 3 + treatment and see if the indices are independent prognostic features.

4. Have the authors tried to combine the indices with each other? Do the same patients have unfavorable values ​​for different indices? Or are these different patients?

5. There are many borrowings in the text, the number of coincidences should be reduced.

Reviewer 2 Report

Comments and Suggestions for Authors

1.         In the results section of the abstract, "At multivariate analysis, age at diagnosis, pretreatment albumin, and squamous histology were significantly associated with OS," it is necessary to specify whether age at diagnosis, pretreatment albumin, and squamous histology are positively or negatively associated with OS.

2.         Previous studies have shown that indicators such as LIPI and SIS are associated with systemic inflammation in lung cancer patients receiving immunotherapy. Please highlight the novelty of this study in the Introduction section.

3.         Some content in Table 1 can be moved to supplementary materials.

4.         In Table 3, the P-value in the LIPI row is not well aligned.

5.         In the figure, the survival curves lack the display of HR, 95% CI, and P-values, which do not meet the standard presentation requirements for survival curves.

6.         When ALI, LIPI, PNI, and SIS indicators are inconsistent, how should these indicators be used to predict outcomes for advanced lung cancer patients receiving immunotherapy? The authors should analyze this in the discussion section. In addition, the authors should further explore the potential reasons why ALI, LIPI, PNI, and SIS can serve as predictive factors.

7. The post-therapy change of cancer tissue is recently reported, such as chemotherapy ( Li et al. DOI: 10.1016/j.canlet.2023.216583) and radiotherapy (Wang et al. DOI: 10.1002/smtd.202200570). In the discussion,authors should include this to assess the prognostic role of these bio-markers.

Comments on the Quality of English Language

The manuscript should be edited by a native English speaker, especially for punctuation and grammar.